# CAT: COMPRESSION-AWARE TRAINING FOR BANDWIDTH REDUCTION

## ABSTRACT

Convolutional neural networks (CNNs) have become the dominant neural network architecture for solving visual processing tasks. One of the major obstacles hindering the ubiquitous use of CNNs for inference is their relatively high memory bandwidth requirements, which can be a main energy consumer and throughput bottleneck in hardware accelerators. Accordingly, an efficient feature map compression method can result in substantial performance gains. Inspired by *quantization-aware training* approaches, we propose a compression-aware training (CAT) method that involves training the model in a way that allows better compression of feature maps during inference. Our method trains the model to achieve low-entropy feature maps, which enables efficient compression at inference time using classical transform coding methods. CAT significantly improves the state-of-the-art results reported for quantization. For example, on ResNet-34 we achieve 73.1% accuracy (0.2% degradation from the baseline) with an average representation of only 1.79 bits per value. Reference implementation accompanies the paper.

## 1 INTRODUCTION

Deep Neural Networks (DNNs) have become a popular choice for a wide range of applications such as computer vision, natural language processing, autonomous cars, etc. Unfortunately, their vast demands for computational resources often prevents their use on power-challenged platforms. The desire for reduced bandwidth and compute requirements of deep learning models has driven research into quantization (Hubara et al., 2016; Yang et al., 2019b; Liu et al., 2019; Gong et al., 2019), pruning (LeCun et al., 1990; Li et al., 2017; Molchanov et al., 2019), and sparsification (Gale et al., 2019; Dettmers & Zettlemoyer, 2019).

In particular, quantization works usually focus on scalar quantization of the feature maps: mapping the activation values to a discrete set $\{q_i\}$ of size $L$. Such a representation, while being less precise, is especially useful in custom hardware, where it allows more efficient computations and reduces the memory bandwidth. In this work, we focus on the latter, which has been shown to dominate the energy footprint of CNN inference on custom hardware (Yang et al., 2017). We show that the quantized activation values $\{q_i\}$ can further be coded to reduce memory requirements.

The raw quantized data require $\lceil \log_2(L) \rceil$ bits per value for storage. This number can be reduced by compressing the feature maps. In particular, in the case of element-wise compression of independent identically distributed values, the lower bound of amount of bits per element is given by the entropy (Shannon, 1948):

$$H(\mathbf{q}) = -\sum_{i=1}^{L} p(q_i) \log_2 p(q_i) \tag{1}$$

of the quantized values $\{q_i\}$, where $p(q_i)$ denotes the probability of $q_i$.

In this work, we take a further step by manipulating the distribution of the quantization values so that the entropy $H(q)$ is minimized. To that end, we formulate the training problem by augmenting the regular task-specific loss (the cross-entropy classifier loss in our case) with the feature map entropy serving as a proxy to the memory rate. The strength of the latter penalty is controlled through a

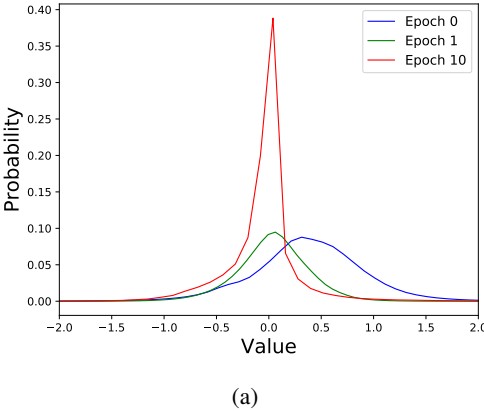 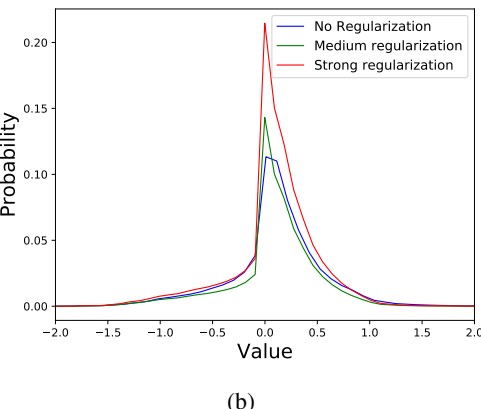

(a)  (b)

Figure 1: Pre-activation distributions of one layer in ResNet-18. **(a) Evolution at different epochs.** As training progresses, the probability of non-positive pre-activation values increases, zeroing more post-ReLU values. The sharp peak at zero reduces entropy and thus improves compressibility. **(b) Effect of entropy regularization.** Without regularization, the distribution has much heavier tails and thus has higher entropy. With increasing regularization, the probability of extreme values is significantly reduced. The entropy penalty $\lambda$ was selected so that the overall accuracy is not affected. The compression ratio in the strongly regularized case is 2.23 higher compared to the unregularized the baseline.

parameter $\lambda > 0$. Fig. 1 demonstrates the effect of the entropy penalty on the compressibility of the intermediate activations.

**Contributions.** Our paper makes several contributions. Firstly, we introduce Compression-Aware Training (CAT), a technique for memory bandwidth reduction. The method works by introducing a loss term that penalizes the entropy of the activations at training time and applying entropy encoding (e.g., Huffman coding) on the resulting activations at inference time. Since the only overhead of the method at inference time is entropy encoding that can be implemented efficiently, the improvement is universal for any hardware implementation, being especially efficient on computationally optimized ones, where memory I/O dominates the energy footprint (Yang et al., 2017; Jouppi et al., 2017). We demonstrate a 2 to 4-fold memory bandwidth reduction for multiple architectures: MobileNetV2 and ResNet on the ImageNet visual recongition task, and SSD512 on the PASCAL VOC object detection task. We also investigate several differentiable loss functions which lead to activation entropy minimization and show a few alternatives that lead to the same effect.

Finally, we analyze the rate-distortion tradeoff of the method, achieving even stronger compression at the expense of minor accuracy reduction: for ResNet-18, we manage to achieve entropy inferior to one bit per value, at the expense of losing 2% of the top-1 accuracy.

## 2 RELATED WORK

Recent studies (Yang et al., 2017; Wang et al., 2019) have shown that almost 70% of the energy footprint on custom hardware is due to the data movement to and from the off-chip memory. Nonetheless, the techniques for memory bandwidth reduction have not yet received significant attention in the literature. One way to improve memory performance is by fusing convolutional layers (Xiao et al., 2017; Xing et al., 2019), reducing the number of feature maps transfers. This reduces both runtime and energy consumption of the hardware accelerator. Another option is to use on-chip cache instead of external memory. Morcel et al. (2019) have shown order of magnitude improvement in power consuption using this technique. Another important system parameter dominated by the memory bandwidth is latency. Jouppi et al. (2017) and Wang et al. (2019) showed that the state-of-the-art DNN accelerators are memory-bound, implying that increasing computation throughput without reducing the memory bandwidth barely affects the total system latency.

Quantization reduces computation and memory requirements; 16-bit fixed point has become a *de facto* standard for fast inference. In most applications, weights and activations can be quantized down to 8 bits without noticeable loss of precision (Lee et al., 2018; Yang et al., 2019a). Further quantization to lower precision requires non-trivial techniques (Mishra et al., 2018; Zhang et al., 2018), which are currently capable of reaching around 3-4 bits per entry without compromising precision (Choi et al., 2018b;a; Dong et al., 2019).

Another way to reduce memory bandwidth is by compressing the intermediate activations prior to their transfer to memory with some computationally cheap encoding, such that Huffman (Chandra, 2018; Chmiel et al., 2019) or run-length (RLE) encoding (Cavigelli et al., 2019). A similar approach of storing only nonzero values was utilized by Lin & Lai (2018). Chmiel et al. (2019) used linear dimensionality reduction (PCA) to increase the effectiveness of Huffman coding, while Gudovskiy et al. (2018) proposed to use nonlinear dimensionality reduction techniques.

Lossless coding was previously utilized in a number of ways for DNN compression: Han et al. (2016) and Zhao et al. (2019) used Huffman coding to compress weights, while Wijayanto et al. (2019) used more complicated DEFLATE (LZ77 + Huffman) algorithm for the same purpose. Aytekin et al. (2019) proposed to use compressibility loss, which induces sparsity and has been shown (empirically) to reduce entropy of the non-zero part of the activations.

## 3 METHOD

We consider a feed-forward DNN $\mathcal{F}$ composed of $L$ layers; each subsequent layer processing the output of the previous one: $x^i = \mathcal{F}_i\big(x^{i-1}\big)$, using the parameters $\mathbf{w} \in \mathbb{R}^N$. We denote by $x^0 = x$ and $x^L = y$ the input and the output of the network, respectively.

The parameters $\mathbf{w}$ of the network are learned by minimizing $\mathcal{L}(x, y; \mathbf{w}) + \lambda \mathcal{R}(\mathbf{w})$, with the former term $\mathcal{L}$ being the task loss, and the latter term $\mathcal{R}$ being a regularizer (e.g., $\|\mathbf{w}\|_2$) inducing some properties on the parameters $\mathbf{w}$.

### 3.1 ENTROPY ENCODING AND RATE REGULARIZATION

Entropy encoders are a family of lossless data encoders which compress each symbol independently. In this case, assuming i.i.d. distribution of the input, it has been show that optimal code length is $-\log_b p$, where $b$ is number of symbols and $p_i$ is the probability of i[th] symbol (Shannon, 1948). Thus, for a discrete random variable $X$ we define an entropy $H(X) = -\mathbb{E}\log_2 X = -\sum_i p(x_i)\log_2 p(x_i)$, which is a lower bound of the amount of information required for lossless compression of $X$. The expected total space required to encode the message is $N \cdot H$, where $N$ is number of symbols. Since we encode the activations with entropy encoder before writing them into memory, we would like to minimize the entropy of the activations to improve the compression rate.

One example of entropy encoder is Huffman coding – a prefix coding which assigns shorter codes to the more probable symbols. It is popular because of simplicity along with high compression rate (Szpankowski, 2000) bounded by $H(X) \le R \le H(X) + 1$ (the comparison of Huffman coding rates to entropy is shown in Fig. 3). Another entropy encoder is arithmetic coding – highly efficient encoder which achieves optimal rates for big enough input but requires more computational resources for encoding.

### 3.2 DIFFERENTIABLE ENTROPY-REDUCING LOSS

Since the empirical entropy is a discrete function, it is not differentiable and thus cannot be directly minimized with gradient descent. Nevertheless, there exist a number of differentiable functions which either approximate entropy or have same minimizer. Thus we optimize

$$\mathcal{L} = \mathcal{L}_{\mathrm{p}} + \lambda \mathcal{L}_H, \tag{2}$$

where $\mathcal{L}_{\mathrm{p}}$ is a target loss function and $\mathcal{L}_H$ is some regularization which minimizes entropy.

**Soft entropy**    First, we consider the differentiable entropy estimation suggested by Agustsson et al. (2017). We start from definition of the entropy,

$$H(X) = -\sum p(x_i) \log(p(x_i)) \tag{3}$$

$$p_i = \frac{|\{x|x = q_i\}|}{N}, \tag{4}$$

where $\mathbf{q}$ is a vector of quantized values. Let $m$ be an index of the bin the current value is mapped to, and $\mathbf{Q}$ a one-hot encoding of this index, i.e.,

$$q_m = \arg\min_{q_i \in \mathcal{Q}} |x - q_i| = \arg\max_{q_i \in \mathcal{Q}} (-|x - q_i|) \tag{5}$$

$$\mathbf{Q}_i = \delta_{im}, \tag{6}$$

where $\delta_{im}$ denotes Kroneker's delta. To make the latter expression differentiable, we can replace argmax with softmax:

$$\tilde{\mathbf{Q}}(x) = \text{softmax}(-|\mathbf{x} - \mathbf{q}|, T), \tag{7}$$

where $T$ is the temperature parameter, and $\tilde{Q}(x) \to Q(x)$ as $T \to \infty$.

Finally, the soft entropy $\hat{H}$ is defined as

$$\hat{H}(X) = -\sum \hat{p}(x_i) \log(\hat{p}(x_i)) \tag{8}$$

$$\hat{p}_i = \frac{\sum_j \tilde{\mathbf{Q}}_i(x_j)}{N} \tag{9}$$

To improve both memory requirements and time complexity of the training, we calculate soft entropy only on part of the batch, reducing the amount of computation and the gradient tensor size. We empirically confirm that this choice gives a reasonable approximation of the real entropy (Appendix B).

**Compressibility loss for entropy reduction**    An alternative loss promoting entropy reduction was proposed by Aytekin et al. (2019) under the name of *compressibility loss* and based on earlier work by Hoyer (2004):

$$\mathcal{L}_c = \frac{\|\mathbf{x}\|_1}{\|\mathbf{x}\|_2} \tag{10}$$

This loss has the advantage of computational simplicity, and has been shown both theoretically and practically to promote sparsity and low entropy in input vectors. While originally applied to the weights of the network, here we apply the same loss to the activations.

As shown in Section 4.1, both the soft entropy and the compressibility loss lead to similar results.

Our method for reducing memory bandwidth can be described as follows: at training time, we fine-tune (training from scratch should also be possible, but we have not tested it) the pre-trained network $\mathcal{F}$ with the regularized loss (2), where we use $\mathcal{L}_H = \sum \hat{H}(x_i)$ in case of differentiable entropy and $\mathcal{L}_H = \sum \mathcal{L}_c(x_i)$ in case of compressability loss. At test time, we apply entropy coding on the activations before writing them to memory, thus reducing the amount of memory transactions. In contrast to Chmiel et al. (2019), who avoided fine-tuning by using test time transformation in order to reduce entropy, our method does not requires complex transformations at test time by inducing low entropy during training.

## 4    EXPERIMENTAL RESULTS

We evaluate the proposed scheme on common CNN architectures for image classification on ImageNet (ResNet-18/34/50, MobileNetV2), and object detection on Pascal VOC dataset (SSD512[1] (Liu et al., 2016)). The weights were initialized with pre-trained model and quantized with uniform quantization

---

[1]Our code is based on implementation by Li (2018).

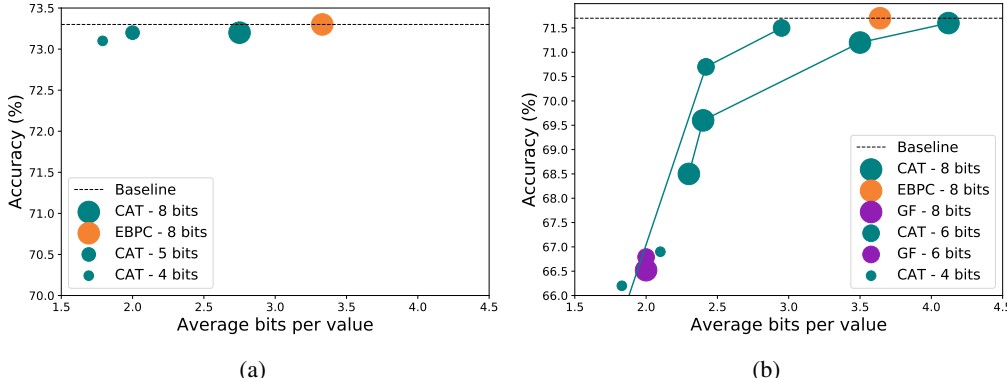

Figure 2: Comparison with other methods: EPBC (Cavigelli et al. (2019)) and GF (Gudovskiy et al. (2018)) in **(a) ResNet-34** and **(b) MobilenetV2**. Different marker size refers to different activation bitwidths before compression. For GF, compression rate was averaged only over compressed layers.

Table 1: Results for ResNet-18, ResNet-50, and SSD512. For comparison we include the results obtained by Gudovskiy et al. (2018) for the SSD512 model on the same task but with a different backbone, for which we obtain a better compression rate with a lower accuracy degradation. Compute denotes activation bitwidth used for arithmetic operations. Memory denotes average number of bits for memory transactions (after compression). Compression ratio denotes the reduction in representation size. Weight bitwidth is 8 except the full-precision experiments. Additional experimental results are provided in Appendix A.

| Architecture | Compute (bits) | Memory (bits) | Compression ratio | Top-1 accuracy (%) |
|---|---|---|---|---|
| ResNet-18, CAT | 32 | 32 | 1 | 69.70 |
| | 5 | 1.5 | 3.33 | 69.20 |
| | 4 | 1.51 | 2.65 | 68.08 |
| ResNet-50, CAT | 32 | 32 | 1 | 76.1 |
| | 5 | 1.60 | 3.125 | 74.90 |
| | 4 | 1.78 | 2.25 | 74.50 |
| SSD512-SqueezeNet (Gudovskiy et al., 2018) | 32 | 32 | 1 | 68.12 |
| | 8 | 2 | 4 | 64.39 |
| | 6 | 2 | 3 | 62.09 |
| SSD512-VGG, CAT | 32 | 32 | 1 | 80.72 |
| | 6 | 2.334 | 2.57 | 77.49 |
| | 4 | 1.562 | 2.56 | 77.43 |

using the shadow weights, i.e. applying updates to a full precision copy of quantized weights (Hubara et al., 2016; Rastegari et al., 2016). The activation were clipped with a learnable parameter and then uniform quantized as suggested by Baskin et al. (2018). Similarly to previous works (Zhou et al., 2016; Rastegari et al., 2016), we used the straight-through estimator (Bengio et al., 2013) to approximate the gradients. We quantize all layers in the network, in contrast to the common practice of leaving first and last layer in high-precision (Zhou et al., 2016; Baskin et al., 2018). Since the weights are quantized only to 8 bit, we have noticed small to no difference between the quantized and non-quantized weights.

For optimization, we use SGD with a learning rate of $10^{-4}$, momentum 0.9, and weight decay $4 \times 10^{-5}$ for up to 30 epochs (usually, $10 - 15$ epochs were sufficient for convergence). Our initial choice of temperature was $T = 10$, which performed well. We tried, following the common approach (Jang et al., 2016), to apply exponential scheduling to the temperature, but it did not have any noticeable effect on the results.

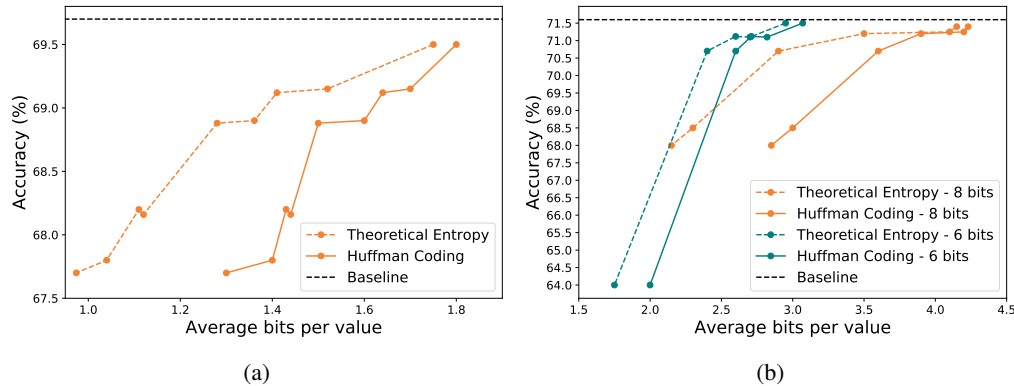

(a)                                                      (b)

Figure 3: Tradeoff between rate and accuracy for different values of λ (ranged between 0 and 0.3) in **(a) ResNet-18** and **(b) MobileNetV2**. In ResNet-18 the activations are quantized to 5 bits, in MobileNet we show results for activation quantized to 6 and 8 bits.

Table 2: Mean and standard deviation over multiple runs of ResNet-18 and ResNet-34.

| Architecture | Compute, bits | Runs | Accuracy, % (mean±std) | Memory, bits (mean±std) |
|---|---|---|---|---|
| ResNet-18 | 5 | 5 | 69.122±0.016 | 1.5150±0.0087 |
| ResNet-34 | 5 | 4 | 73.025±0.095 | 1.7875±0.033 |

In Fig. 2 we compare our method with EPBC (Cavigelli et al. (2019)) and GF (Gudovskiy et al. (2018)). EPBC is based on a lossless compression method that maintains the full precision accuracy while reducing bit rate to approximately 3.5 bits/value in both models. GF, on the other hand, provides strong compression at the expense of larger accuracy degradations. In addition, Gudovskiy et al. (2018) have compressed only part of the layers. Unlike these two methods, CAT allows more flexible tradeoff between compression and accuracy. CAT shows better results in ResNet-34 and show either better accuracy or compression for MobileNetV2. We also run our method on additional architectures: ResNet-18, ResNet-50, and SSD512 with VGG backbone; the results are listed in Table 1. Even though we can not directly compare detection results with Gudovskiy et al. (2018), the drop in accuracy is lower in our case.

## 4.1 ABLATION STUDY

**Rate-Accuracy Tradeoff**    The proposed CAT algorithm tries to balance between minimizing the rate and maximizing the accuracy of the network by means of the parameter λ in Eq. (2). To check this tradeoff, we run the same experiment in ResNet-18 and MobileNetV2 with different values of λ in the range of $0 - 0.3$, with results shown in Fig. 3. We show the results of the theoretical entropy and of the Huffman coding, which was chosen for its simplicity but more efficient encoders can be combined with our method. For higher rate Huffman coding is close (∼3% overhead) to the theoretical entropy, while for lower entropy the difference is higher and is bounded by 1 bit per value – in the latter case, different lossless coding schemes such as arithmetic coding can provide better results.

**Robustness**    For checking the robustness of our method, we performed several runs with the same hyper-parameters and a different random seed. Statistics reported in Table 2 suggest that the method is robust and stable under random intialization.

**Soft entropy vs. compressibility loss**    Replacing the soft entropy with a different loss which minimizes the entropy almost did not affect the results, as shown in Table 3. We conclude that the desired effect is a result of entropy reduction rather than a particular form of regularization promoting it.

Table 3: Performance of soft entropy (8) and compressibility loss (10) on ResNet-18.

| Compute, bits | Loss | Accuracy | Memory, bits |
|:---:|:---:|:---:|:---:|
| 4 | entropy | 67.86% | 1.43 |
| 4 | compressability | 67.84% | 1.50 |
| 5 | entropy | 69.49% | 1.79 |
| 5 | compressability | 69.36% | 1.73 |

**Batch size**    We have noticed that training ResNet-50 on a single GPU mandated the use of small batches, leading to performance degradation. Increasing the batch size from 16 to 64 without changing other hyperparameters increased accuracy by more than 0.5% with entropy increase by less than 0.1 bits/value.

## 5    DISCUSSION

Quantization of activations reduces memory access costs which are responsible for a significant part of energy footprint in NN accelerators. Conservative quantization approaches, known as post-training quantization, take a model trained for full precision and directly quantize it to 8-bit precision. These methods are simple to use and allow for quantization with limited data. Unfortunately, post-training quantization below 8 bits usually incurs significant accuracy degradation. Quantization-aware training approaches involve some sort of training either from scratch (Hubara et al., 2016), or as a fine-tuning step from a pre-trained floating point model (Han et al., 2016). Training usually compensates significantly for model's accuracy loss due to quantization.

In this work, we take a further step and propose a compression-aware training method to aggressively compress activations to as low as 2 average bit/value representations without harming accuracy. Our method optimizes the average-bit per value needed to represent activation values by minimizing the entropy. We demonstrate the applicability of our approach on classification tasks using the models MobileNetV2, ResNet, and object detection task using the model SSD512. Due to the low overhead, the method provides universal improvement for any custom hardware, being especially useful for the accelerators with efficient computations, where memory transfers are a significant part of the energy budget. We show that the effect is universal among loss functions and robust to random initialization.

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

# A    ADDITIONAL RESULTS

We list results of the additional experiments we performed in Tables A.1 and A.2.

Table A.1: Experimental results for ResNet.

| Architecture | Batch size | lr | $\lambda$ | Compute | Memory | Accuracy, % |
|---|---|---|---|---|---|---|
| ResNet-18 | 96 | 0.001 | 0 | 4 | 2.050 | 68.000 |
| | | | 0.05 | | 1.540 | 67.950 |
| | | | 0.08 | | 1.430 | 67.860 |
| | | | 0.05 | 5 | 1.790 | 69.490 |
| | | | 0.05 | | 1.750 | 69.400 |
| | | | 0.1 | | 1.410 | 69.120 |
| | | | 0.12 | | 1.361 | 68.900 |
| | | | 0.15 | | 1.280 | 68.914 |
| | | | 0.18 | | 1.120 | 68.160 |
| | | | 0.2 | | 1.110 | 68.300 |
| | | | 0.25 | | 1.040 | 67.800 |
| | | | 0.3 | | 0.974 | 67.700 |
| | | | 0 | 6 | 3.100 | 70.000 |
| | | | 0.05 | | 1.930 | 69.710 |
| | | | 0.08 | | 1.700 | 69.500 |
| | | | 0.05 | 7 | 2.280 | 69.660 |
| | | | 0 | 8 | 5.100 | 69.900 |
| | | | 0.05 | | 2.460 | 69.820 |
| | | | 0.08 | | 2.410 | 69.110 |
| ResNet-34 | 96 | 0.001 | 0.05 | 8 | 2.750 | 73.200 |
| | | | 0.05 | 6 | 2.000 | 73.200 |
| | | | 0.05 | 5 | 1.790 | 73.100 |
| ResNet-50 | 16 | 0.0001 | 0 | 4 | 2.500 | 73.700 |
| | 16 | | 0.05 | | 1.720 | 73.800 |
| | 64 | | 0.05 | | 1.78 | 74.5 |
| | 48 | | 0.08 | | 1.67 | 74.2 |
| | 16 | | 0 | 5 | 2.950 | 75.500 |
| | 16 | | 0.05 | | 1.920 | 75.460 |
| | 16 | | 0.08 | | 1.700 | 75.200 |
| | 16 | | 0.1 | | 1.600 | 74.900 |

# B    SAMPLE SIZE IN SOFT ENTROPY CALCULATION

To check whether the amount of values used to calculate soft entropy is enough, we ran a soft entropy evaluation on a single tensor and compared it to real values. Since the tensors are large (hundreds of thousands of elements), even 5% of the values already provides reasonable approximation of the real entropy, as shown in Fig. B.1.

Table A.2: Experimental results for MobileNet.

| Architecture | Batch size | lr | $\lambda$ | Compute | Memory | Accuracy, % |
|---|---|---|---|---|---|---|
| | 64 | 0.0001 | 0 | | 2.200 | 66.150 |
| | 64 | 0.001 | 0 | | 2.800 | 66.200 |
| | 64 | 0.0001 | 0.05 | | 2.100 | 66.900 |
| | 64 | 0.001 | 0.05 | 4 | 2.080 | 66.400 |
| | 64 | 0.001 | 0.08 | | 1.830 | 66.200 |
| | 64 | 0.0001 | 0.08 | | 1.980 | 66.450 |
| | 64 | 0.001 | 0 | | 3.900 | 69.600 |
| | 64 | 0.0001 | 0 | | 3.700 | 71.000 |
| | 96 | 0.0001 | 0.05 | | 2.950 | 71.500 |
| | 32 | 0.0001 | 0.08 | 6 | 2.700 | 70.930 |
| | 64 | 0.0001 | 0.1 | | 2.700 | 70.950 |
| MobileNetV2 | 32 | 0.001 | 0.15 | | 1.750 | 64.000 |
| | 64 | 0.0001 | 0.15 | | 2.600 | 71.200 |
| | 64 | 0.0001 | 0.2 | | 2.450 | 70.700 |
| | 64 | 0.0001 | 0 | | 4.750 | 71.300 |
| | 64 | 0.0001 | 0.05 | | 4.150 | 71.400 |
| | 64 | 0.001 | 0.08 | | 2.600 | 70.000 |
| | 64 | 0.0001 | 0.08 | | 4.120 | 71.600 |
| | 64 | 0.001 | 0.1 | 8 | 2.400 | 69.600 |
| | 64 | 0.0001 | 0.1 | | 4.100 | 71.250 |
| | 64 | 0.001 | 0.15 | | 2.300 | 68.500 |
| | 64 | 0.001 | 0.2 | | 2.150 | 68.000 |
| | 64 | 0.0001 | 0.2 | | 3.500 | 71.200 |
| | 32 | 0.0001 | 0.3 | | 2.900 | 70.700 |

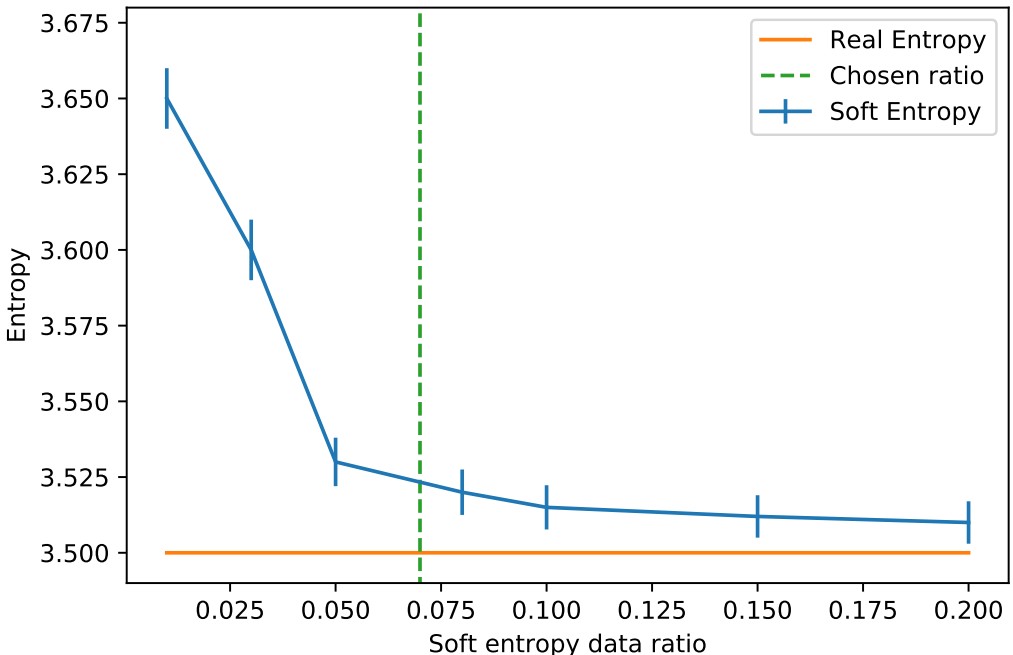

Figure B.1: Soft entropy with different sample size compared to a real entropy of a single batch. Error bars are for standard deviation over 3 runs.

