# OpenReview forum: "CAT: Compression-Aware Training for bandwidth reduction"
_ICLR.cc/2020/Conference — Reject_

### Official Review · AnonReviewer1 · 2019-10-18
**Official Blind Review #1**

**Rating:** 6

**Review:**

Summary:
The authors propose a method for training easy-to-quantize models that are quantized after training (post-training quantization). They do so by regularizing by the entropy, thereby forcing the weight distribution to be more compressible. They further compress the weights using entropy coding.

Strengths of the paper:
- The paper presents strong experimental results on ResNet, SqueezeNet VGG and MobileNet architectures and provides the code, which looks sensible.

Weaknesses of the paper:
- The authors could have applied CAT to other tasks such as Image Detection, while proving inference times on CPUs. Indeed, it is unclear to me what would be the influence of the entropic decoder which is claimed to be fast for "efficient implementations" by the authors.
- The idea of regularizing by the entropy is not novel (see for instance "Entropic Regularization", Grandvalet et a.l),  as well as the idea of further encoding the weights using entropic coders (as in "Deep Compression", Han et al.).

Justification of rating:
The authors present an intuitive method (yet not novel) for quantizing the weights of a neural network. My main concern would be about the inference time but I consider that the experimental results suggest strong evidence that CAT performs well on a wide variety of architectures.

**Experience Assessment:**

I have published one or two papers in this area.

**Review Assessment: Checking Correctness Of Derivations And Theory:**

N/A

**Review Assessment: Checking Correctness Of Experiments:**

I carefully checked the experiments.

**Review Assessment: Thoroughness In Paper Reading:**

I read the paper thoroughly.

---

> ### Author Response · Authors · 2019-11-06
> **Answer to reviewer #1**
>
> Thank you for your review and your comments. We would like to provide an answer to the issues raised in your review.
>
> Q: The authors could have applied CAT to other tasks such as Image Detection while proving inference times on CPUs. Indeed, it is unclear to me what would be the influence of the entropic decoder which is claimed to be fast for "efficient implementations" by the authors.
> A: We have applied the CAT to object detection task (SSD512, Table 3). As we have written in the answer to Reviewer #3, even for naive implementation of Huffman coding, the overhead for GPU-running code is approx. 3-4%. When considering dedicated hardware implementations, the overhead can be made negligible.
>
> Q: The idea of regularizing by the entropy is not novel (see for instance "Entropic Regularization", Grandvalet et al.), as well as the idea of further encoding the weights using entropic coders (as in "Deep Compression", Han et al.).
> A: The idea of using entropy regularization, in general, is indeed not novel and we mention relevant works in the Related Work section. However, using a differentiable entropy approximation as a loss term to improve the compressibility of the activations is novel.

---

### Official Review · AnonReviewer4 · 2019-11-04
**Official Blind Review #4**

**Rating:** 6

**Review:**

The format of the paper does not meet the requirement of ICLR. Due to this, I will give a 3. I suggest the authors to change it as soon as possible.

Besides that, the main idea of the paper is to regularize the training of a neural network to reduce the entropy of its activations. There are extensive experiments in the paper.

The paper introduce two kinds of method to regularize the entropy. The first method is a soft version of the original entropy, and the second is the compressibility loss. After adding the regularization, the performance drop of the compressed network is reduced. The experiment performance is promising.

I think the method is straightforward and reasonable with only a few questions:
1. Why do you quantize the weight? Seems it's not necessary because the paper only address activation quantization.
2. What will happen if the weights are quantized to lower bits? For example, 4bit?
2. How about adding the regularization to weights?


**Experience Assessment:**

I have read many papers in this area.

**Review Assessment: Checking Correctness Of Derivations And Theory:**

I assessed the sensibility of the derivations and theory.

**Review Assessment: Checking Correctness Of Experiments:**

I assessed the sensibility of the experiments.

**Review Assessment: Thoroughness In Paper Reading:**

I read the paper thoroughly.

---

> ### Author Response · Authors · 2019-11-05
> **Format fix**
>
> Thank you for your review!
>
> As proposed, we have updated the pdf with the right format. For some reason, one of TeX packages interfered with it. We are going to address other points of your review later.

---

> ### Author Response · Authors · 2019-11-06
> **Answer to reviewer #4**
>
> Once again, thank you for your review and your comments. We will answer the questions in the review.
>
> Q: Why do you quantize the weight? Seems it's not necessary because the paper only addresses activation quantization.
> A: While our paper focuses on activation quantization, one of the goals of our research was an actual hardware implementation of the neural network, which is significantly more efficient when the weights are quantized. We, therefore, decided to demonstrate the combined effect of weight and activation quantization; however, in our experiments, we also show limit cases of 32-bit weights which do not impact the general picture.
>
>
> Q: What will happen if the weights are quantized to lower bits? For example, 4bit?
> A: We believe that our method can be used along with methods of weight quantization only minor impact of the results. However, adding a complicated weight quantization method to CAT would require more resources without contributing much to the analysis of CAT. Thus, we have chosen a simple quantization method that does not achieve baseline accuracy at lower bits.
>
> Q: How about adding the regularization to weights?
> A: Our method could also be applied to the weights. In this paper, we focus on activations, since the benefit of the lossless compression activation is more significant due to the fact they are usually responsible for most of the memory accesses.
> In addition, the paper by Aytekin et al. proposed compressibility loss and has successfully applied it to the weights.

---

> > ### Comment · AnonReviewer4 · 2019-11-11
> > **Thank You for Your Response**
> >
> > I appreciate the quick response, and I find them reasonable enough. I will change the score.

---

### Official Review · AnonReviewer3 · 2019-11-05
**Official Blind Review #3**

**Rating:** 6

**Review:**

In this article, the authors propose a compression-aware method to achieve efficient compression of feature maps during the inference procedure. When in the training step, the authors introduce a regularization term to the target loss that optimizes the entropy of the activation values. At inference time, an entropy encoding method is applied to compress the activations before writing them into memory. The experimental results indicate that the proposed method achieves better compression than other methods while holding accuracy.
There are still some issues as follows:
1.	The authors should carefully check the format of the references in the whole article. For example, in section 2, line 5 from the top and line 8 from the bottom, “Xiao et al. (2017), Xing et al. (2019)” and “(Chmiel et al., 2019)” are in the wrong format.
2.	It is suggested that the authors swap the order of formulation (8) and (9) in section 3.2 so that it will be a good correlation with the formulation (3) and (4).
3.	I am interested in learning the time taken by the proposed method during the inference procedure vs other related methods.
4.	The authors studied two differentiable entropy approximation in the paper, and they stated that they calculate soft entropy only on the part of the batch for the reduction of both memory requirements and time complexity in training. I hope the authors will clarify 1) Whether the accuracy will be affected by other differentiable entropy approximations; 2) what is the impact on accuracy if only part of the batch is considered.


**Experience Assessment:**

I have published one or two papers in this area.

**Review Assessment: Checking Correctness Of Derivations And Theory:**

I assessed the sensibility of the derivations and theory.

**Review Assessment: Checking Correctness Of Experiments:**

I assessed the sensibility of the experiments.

**Review Assessment: Thoroughness In Paper Reading:**

I read the paper thoroughly.

---

> ### Author Response · Authors · 2019-11-06
> **Answer to reviewer #3**
>
> Thank you for your review and your comments. We fixed the wrong reference format and switched the equations as proposed. We also would like to provide an answer to the issues raised in your review.
>
> Q: I am interested in learning the time taken by the proposed method during the inference procedure vs other related methods.
> A: In inference time, our method requires only applying an entropy encoder, for ResNet-18 we measured a 3-4% overhead for building a Huffman tree with our naive implementation. More efficient implementations would probably make the difference totally negligible. We can further reduce the overhead by fixing the tree which will make it even faster. Mapping the values is in fact as fast as fetching them from the main memory. We believe, lower level implementation should yield latency improvements due to faster I/O with main memory.
>
> Q:  Whether the accuracy will be affected by other differentiable entropy approximations?
> A: Entropy approximation does not affect accuracy directly. The accuracy decrease is a result of an additional loss term. By changing the relative weight of the entropy term, we can reach different accuracy/compression ratios, as shown in Fig. 3. However, for the same model accuracy, the better is the entropy approximation, the better compression will be attained.
>
> Q:  what is the impact on accuracy if only part of the batch is considered.
> A: Since there is only minor difference between the two methods (soft entropy based only on part of the tensor and compressibility loss which is calculated on the whole tensor), we believe there is no significant impact of using only part of the batch. Since the tensors are large, the amount of values used for approximation is still relatively big. To check this assumption, we ran a soft entropy evaluation on a single tensor and compared it to real values, and added the results to Appendix B.

---

### Decision · Program_Chairs · 2019-12-19

**Decision:**

Reject

**Comment:**

This work propose a compression-aware training (CAT) method to allows efficient compression of  feature maps during inference. I read the paper myself. The proposed method is quite straightforward and looks incremental compared with existing approaches based on entropy regularization.